# TRAJECTORY-AWARE VERBALIZED OPTIMIZATION FOR MULTI-AGENT SYSTEMS

## ABSTRACT

Large language model (LLM)-based multi-agent systems have shown significant potential, but their effectiveness often depends on manually engineered prompts, which are refined through labor-intensive trial and error. While automatic optimization methods exist, they often rely on coarse, task-level outcomes, neglecting the rich trajectory-level information that captures how agents reason, coordinate, and fail. To address this gap, we propose a **T**rajectory-**A**ware **V**erbalized **O**ptimization (TAVO) framework for prompt refinement in multi-agent systems. Inspired by reinforcement learning, TAVO introduces a credit assignment mechanism that decomposes interaction trajectories into sub-trajectories, linking specific reasoning and coordination steps to the final outcome. This generates fine-grained, process-level feedback. By modeling prompts as verbalized policies, TAVO translates this trajectory feedback into concrete editing instructions, which are aggregated across tasks for systematic refinement. Experiments on both collaborative and competitive multi-agent benchmarks demonstrate that our framework enhances system performance while reducing coordination costs, underscoring the value of leveraging trajectory-level signals to construct more adaptive and efficient LLM-based multi-agent systems.

## 1 INTRODUCTION

The rise of large language models (LLMs) (Brown et al., 2020) has transformed artificial intelligence from standalone systems into ecosystems of interacting autonomous agents. These LLM-based multi-agent systems (Guo et al., 2024), where agents perceive complex environments (Li et al., 2024b) and coordinate through communication to solve tasks beyond individual capabilities (Zhu et al., 2025), are demonstrating broad potential, from collaborative coding in software engineering (Qian et al., 2023) to diagnostic teamwork in healthcare (Li et al., 2024a). At the core of these systems lies a fragile but decisive element: the prompts that specify agent roles, behaviors, and coordination protocols. Prompt quality determines whether the system operates as a cohesive unit or descends into misalignment.

Currently, the craft of prompt engineering remains more of an art than a science, representing a significant bottleneck in the reliable deployment of these systems. Expert practitioners refine prompts through an iterative process of analyzing interaction histories, such as reasoning chains, message exchanges, and coordination patterns, to identify errors and attribute to the specific part of prompt for agents. While effective, this process is labor-intensive and does not scale, especially as the number of agents and interaction complexity grow. To automate this process, recent work has proposed prompt optimization techniques (Khattab et al., 2024; Yuksekgonul et al., 2025), which primarily use final task outcomes as optimization signals. However, the reward on the final outcome is not always enough to reflect the real quality of the multi-agent system (Zhuge et al., 2025). In multi-agent settings, outcomes are the cumulative result of many interdependent steps. A positive reward cannot reflect the potential redundant steps in the whole process (Wu et al., 2025), while a negative reward does not reflect the reason for the error (Zhang et al., 2025). This "black-box" perspective overlooks the trajectory-level evidence that leads to the success or failure. Without examining these steps, optimization can only indicate failure, not diagnose its source or cause. As a result, adjustments are often blunt and generic, failing to resolve specific coordination issues and sometimes introducing inefficiencies (Cemri et al., 2025).

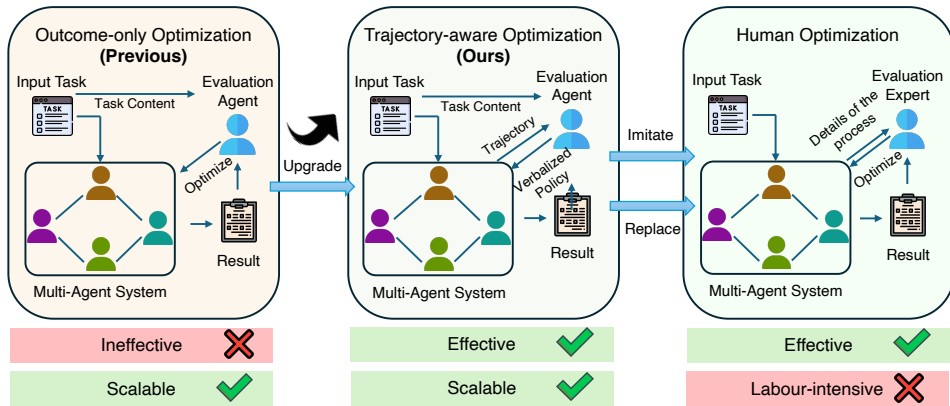

Figure 1: Motivation for TAVO. We identify the key difference in utilizing trajectory information between the current automatic prompt refinement methods and human expert in refining prompt in multi-agent system. Our TAVO aims to fill this gap, via imitating human experts.

We argue that the next breakthrough in multi-agent prompt optimization requires bridging the gap between scalable automation and expert-level insight. The key challenge is to systematically exploit trajectory information. Reinforcement learning (RL) provides a natural pathway, as it assigns credit from outcomes back to individual steps using estimates such as value functions (Jia & Zhou, 2022) or policy advantages (Schulman et al., 2015). RL approaches provide a strong foundation, yet their direct application to multi-agent prompt optimization in natural language introduces open questions regarding efficiency and interpretability in trajectory-level analysis.

To this end, we introduce the **T**rajectory-**A**ware **V**erbalized **O**ptimization (TAVO) framework (Figure 1), a methodology designed to emulate the diagnostic process of human experts: learning from the entire interaction trajectory rather than only the final outcome. TAVO builds on two synergistic pillars, inspired by RL. First, it incorporates a trajectory-aware credit assignment mechanism, to decompose long, complex interactions into meaningful sub-sequences and attribute outcomes back to critical segments. This resolves the credit assignment problem in long-horizon tasks, enabling fine-grained, process-level feedback that highlights strengths and weaknesses in coordination. Second, by conceptualizing prompts as verbalized policies—explicit, interpretable specifications—TAVO bridges analytical feedback with actionable intervention. This design allows trajectory-level signals to be translated directly into natural language suggestions for policy refinement, yielding systematic, interpretable, and iterative improvement of multi-agent directives.

Our contributions are substantiated through both theoretical design and empirical validation: (1) We propose TAVO, a holistic framework that integrates trajectory-level analysis into the automatic prompt optimization loop for multi-agent systems, bridging the gap between expert-inspired debugging and scalable automation. (2) Within TAVO, we introduce a trajectory-aware credit assignment mechanism that delivers fine-grained, localized feedback by linking sub-trajectories to outcomes and distilling clear signals from extended interactions. (3) We further conceptualize prompts as optimizable verbalized policies and generate explicit, natural-language policy-editing instructions from trajectory feedback, enabling systematic and interpretable refinement. (4) We empirically validate TAVO across both collaborative and competitive settings, demonstrating significant performance gains and reduced coordination overhead, thereby highlighting its potential to build more robust and efficient autonomous systems

## 2 RELATED WORKS

### 2.1 LLM-BASED MULTI-AGENT SYSTEM

LLM-based multi-agent systems have emerged as a promising paradigm for complex tasks that require specialization and collaboration. Prior work generally follows two directions. One designs structured role-play workflows, where agents assume distinct roles (e.g., programmer and tester) to mimic real-world processes in domains like code generation (Qian et al., 2023) and medical diag-

nosis (Li et al., 2024a). The other emphasizes dynamic communication strategies, such as debates (Khan et al., 2024) or iterative self-reflection (Madaan et al., 2023), to refine collective reasoning. Despite their effectiveness, both paradigms rely heavily on hand-crafted prompts to specify roles and coordination patterns. Designing and refining these prompts is labor-intensive and does not scale, creating a key bottleneck. To address this, recent work has begun to automate optimization. Program-search methods employ evolutionary algorithms or Monte Carlo Tree Search to discover effective workflows and prompts (Hu et al., 2024; Zhang et al., 2024); gradient-based approaches model the system as a differentiable graph and propagate textual gradients (Yin & Wang, 2025); and staged methods strengthen role prompts before joint tuning (Zhou et al., 2025). These approaches improve robustness, but their reliance on outcome-level feedback provides limited diagnostic insight into coordination failures. Our work departs from this outcome-centric view by incorporating process-level feedback to guide optimization.

## 2.2 PROMPT OPTIMIZATION FOR LLMS

Prompt optimization has evolved from early gradient-based tuning methods (Lester et al., 2021) to approaches that use LLMs themselves as optimizers. One line of work relies on numeric feedback, framing prompt optimization as program synthesis guided by evaluation metrics (Yang et al., 2024; Fernando et al., 2024; Khattab et al., 2024). Another leverages verbal feedback, where natural language critiques and textual gradients provide more nuanced updates (Shinn et al., 2023; Yuksekgonul et al., 2025). While effective in single-agent contexts, these methods primarily depend on outcome-level signals and overlook the process-level information that is crucial in multi-agent coordination. A complementary direction distills prior experiences into reusable natural-language policies. ExpeL aggregates successful trajectories into textual insights retrievable at inference time (Zhao et al., 2024), AutoGuide generates state-conditioned guidelines via contrastive analysis (Fu et al., 2024), and Mobile-AgentE operationalizes shortcuts into executable procedures (Wang et al., 2025). GiGPO further provides fine-grained credit assignment within trajectories (Feng et al., 2025). These works highlight the potential of experience-based policies but remain designed for single-agent scenarios. Our work extends this paradigm by transforming multi-agent trajectories into role-aware policies that explicitly govern interaction and division of labor.

## 2.3 AGENTIC EVALUATION AND THE ROLE OF TRAJECTORIES

Evaluating LLM-based agents is challenging because their problem-solving unfolds as multi-step process (Yao et al., 2023). Existing benchmarks, such as SWE-Bench for software engineering (Jimenez et al., 2024), typically emphasize final success rates. However, outcome-centric evaluation is often an unreliable proxy for capability (Zhuge et al., 2025), as it overlooks robustness, efficiency, and coordination quality. Recent work increasingly recognizes the value of trajectory-level analysis. Trajectories capture reasoning chains, social dynamics, and failure modes (Lù et al., 2025), enabling error attribution (Zhang et al., 2025) and root-cause analysis (Cemri et al., 2025). Building on this perspective, our framework leverages trajectory-level signals as a more informative and reliable basis for optimizing multi-agent systems.

## 3 PRELIMINARY

### 3.1 MULTI-AGENT SYSTEM AS A MARKOV DECISION PROCESS

We formalize an LLM-based multi-agent system as a *Markov game*, a multi-agent extension of the Markov decision process (MDP) framework (Littman, 1994). Consider a task described by $\mathcal{T}$ and a set of agents $\mathcal{A} = \{a_i\}_{i=1}^N$, where each agent $a_i$ is governed by a textual prompt $P_i$ that defines its behavior and role. Formally, the system is defined by the tuple $(\mathcal{S}, \mathcal{A}_i, \mathcal{P}, \mathcal{R}, \gamma)$. **State Space** $\mathcal{S}$: A state $s_t \in \mathcal{S}$ at step $t$ consists of the accumulated dialogue history $\mathcal{H}_{t-1}$, which includes the task description $\mathcal{T}$ and all messages exchanged among agents up to step $t-1$. **Action Space** $\mathcal{A}_i$: Each agent $a_i$ takes an action by generating a message $m_{i,t}$ at step $t$. The joint action of all agents is denoted as $m_t = \{m_{i,t}\}_{i=1}^N$. **Transition Dynamics** $\mathcal{P}$: The state transition is deterministic: given the joint action $m_t$, the new state is updated by appending the new messages to the history, i.e., $s_{t+1} = \mathcal{H}_t = \mathcal{H}_{t-1} \cup \{m_t\}$. **Reward Function** $\mathcal{R}$: After the interaction concludes with a

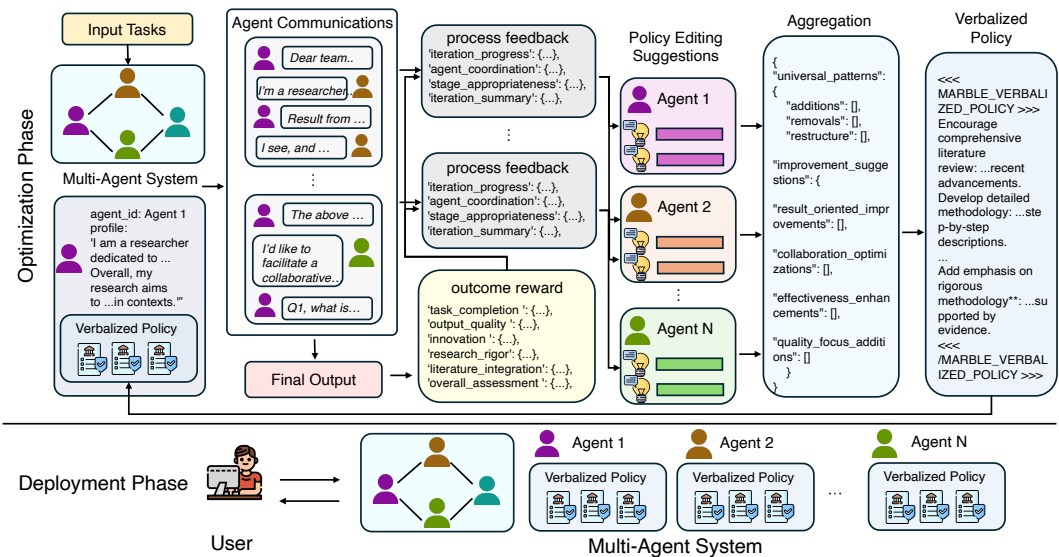

Figure 2: **Overview of our TAVO**. Our framework models the agent's prompt as verbalized policies. It first generates global feedback on the final output, then assigns credit to each sub-trajectory to produce local feedback and corresponding policy-editing suggestions. These suggestions are aggregated by identifying common revisions to refine the prompt. The optimization process is applied iteratively on training tasks, after which the optimized system is deployed to handle unseen tasks.

trajectory $\mathcal{H}_T$ and a final outcome $r$, an evaluation function provides a global outcome feedback signal $R(\mathcal{T}, r)$. **Discount Factor** $\gamma$: This factor determines the present value of future rewards.

Within this formulation, the prompt $P_i$ of each agent functions as a *verbalized policy* $\pi_i^{\text{verbal}}$, which directly conditions the agent's response generation:

$$m_{i,t} \sim \pi_i^{\text{verbal}}(\cdot \mid s_t, P_i). \tag{1}$$

## 3.2 CREDIT ASSIGNMENT IN SEQUENTIAL DECISION MAKING

Reinforcement learning (RL) provides a general framework for estimating process rewards in long-horizon decision making (Sutton et al., 1998) for updating policy. A fundamental challenge in RL is the *credit assignment problem*: rewards are often sparse and delayed, making it difficult to trace the final outcome back to the intermediate actions that most contributed to success or failure. To address this challenge, RL introduces intermediate feedback signals that approximate the long-term contribution of each action. Techniques such as value estimation (Jia & Zhou, 2022), temporal-difference (Sutton, 1988), and advantage functions (Schulman et al., 2015), serve as surrogates for the eventual outcome, enabling agents to update their policies even before the final reward is observed. These process-level signals provide a bridge between sparse outcome feedback and fine-grained policy optimization.

However, a fundamental limitation arises when applying these RL principles to LLM-based multi-agent systems where policies are encoded as textual prompts, especially when the underlying LLM parameters are frozen. Standard RL algorithms are designed for continuous parameter spaces amenable to gradient-based optimization. In contrast, prompt-based policies are discrete, structured in natural language, and cannot be directly optimized via numeric gradients. Consequently, while the conceptual framework of credit assignment is highly relevant, the technical machinery of RL cannot be directly transferred to the problem of optimization of LLM-based multi-agent systems.

## 4 TRAJECTORY-AWARE VERBALIZED OPTIMIZATION

We introduce **Trajectory-Aware Verbalized Optimization (TAVO)**, a framework that refines agent prompts by leveraging the MDP formulation of multi-agent interactions. Within this view, prompts

function as policies, interaction histories form state-action sequences, and optimization is achieved through structured credit assignment and iterative policy improvement. As illustrated in Figure 2, TAVO consists of three core components: (1) **Trajectory-aware credit assignment**, which decomposes sparse outcome-level rewards into fine-grained, verbalized process feedback; (2) **Policy-editing suggestion generation**, which translates this feedback into interpretable, natural language instructions for prompt modification; and (3) **Aggregated policy refinement**, which consolidates suggestions across multiple tasks to identify robust, generalizable improvements.

## 4.1 PROMPTS AS VERBALIZED POLICY

Within the MDP framework, each agent $a_i$ is governed by a verbalized policy $\pi_i^{\text{verbal}}$ instantiated by its prompt $P_i$. At step $t$, the agent generates messages $m_{i,t}$ according to Equation 1. The prompt $P_i$ thus encodes the strategies, rules, and behavioral guidelines that condition the agent's responses. This formulation is particularly critical when using frozen base models, where the prompt serves as the primary mechanism for steering behavior due to constraints on model access or computational resources. The joint trajectory $\mathcal{H}_T$ emerges from the execution of the joint policy $\{\pi_i^{\text{verbal}}\}_{i=1}^N$ over the sequence of states. The final output $r$ is derived from $\mathcal{H}_T$, and the reward function evaluates the task-result pair $(\mathcal{T}, r)$. In this formulation, the main challenge of credit assignment in RL remains.

Under this formulation, we confront the classic credit assignment challenge from RL in a novel context: it is difficult to (1) identify which specific intermediate actions within a long interaction sequence contributed most significantly to the final outcome, and (2) associate that contribution back to specific components of the verbalized policy (the prompt) to guide optimization.

## 4.2 TRAJECTORY-AWARE CREDIT ASSIGNMENT

To address the credit assignment challenge, we propose a *trajectory-aware credit assignment mechanism*. The core insight is that in long-horizon, multi-agent interactions, the trajectory, i.e., the sequence of intermediate responses and decision points, encodes rich information about how the final outcome emerged. Leveraging this information allows for a more precise attribution of credit than relying solely on the final outcome. However, utilizing full trajectories directly is challenging. As task complexity and the number of agents increase, trajectories become lengthy and information-dense. LLMs are known to struggle with such extended contexts, often overlooking critical details while being distracted by irrelevant information (Zhao et al., 2025). Simply feeding the raw trajectory to an LLM risks losing the very signals needed for accurate credit assignment.

Our mechanism overcomes this by first evaluating the final outcome $r$ to obtain a verbalized reward:

$$F_r = R(\mathcal{T}, r), \tag{2}$$

where $R$ may be implemented using either pre-defined rule-based evaluation or LLM-based evaluator, such as LLM-as-a-Judge (Gu et al., 2024) to provide outcome-level feedback. To manage context length, the full trajectory $\mathcal{H}_T$ is decomposed into manageable sub-trajectories $h_t$, each capturing a coherent segment of the interaction (e.g., one round of a debate (Khan et al., 2024) or a single stage in a structured workflow (Qian et al., 2023)). For each sub-trajectory, we derive process feedback by conditioning the outcome evaluation on the partial history:

$$F_t = \text{TAVO}_{\text{credit}}(\mathcal{T}, h_t, F_r), \tag{3}$$

where $\text{TAVO}_{\text{credit}}$ employs an LLM to verbally articulate the contribution of sub-trajectory $h_t$ to the final outcome $F_r$. This process can be executed in parallel for all sub-trajectories, making it efficient regardless of total trajectory length.

Conceptually, this mirrors value decomposition methods in multi-agent reinforcement learning. However, instead of producing numeric estimates, it generates interpretable, verbal signals that attribute credit to specific interaction segments, thereby transforming a sparse outcome evaluation into actionable, step-level guidance.

## 4.3 GENERATING POLICY-EDITING SUGGESTIONS

Even with step-level feedback from trajectory-aware credit assignment, a key challenge remains: determining which agent's policy, and which specific part of its prompt, should be updated based on

that feedback. Prompts are often high-level and semantically entangled, making the link between a trajectory segment and a specific policy component non-obvious.

To bridge this gap, we model prompts as *verbalized policies*, where the prompt consists of a list of explicit policies. The policies can also be customized based on the system, such as a list of rules, instructions, or guidelines that agents should follow. Building on the outcome feedback $F_r$ and step-level feedback $F_t$, TAVO generates structured *policy-editing suggestions* that map verbalized credit to specific prompts. For an agent $a_i$ active in a sub-trajectory $h_t$, edits are proposed as:

$$\Delta P_{i,t} = \text{TAVO}_{\text{suggestion}}(\mathcal{T}, F_r, F_t, P_i), \tag{4}$$

where $\Delta P_{i,t}$ is a set of natural language instructions generated by an LLM (i.e., $\text{TAVO}_{\text{suggestion}}$) to refine prompt $P_i$. $\Delta P_{i,t}$ are structured into three actionable types: (1) **Addition**: introducing new rules that encourage effective behaviors observed in the trajectory; (2) **Deletion**: removing policies that led to harmful actions; and (3) **Modification**: rephrasing existing instructions to better align with task goals. This process is analogous to calculating a policy gradient in reinforcement learning. However, instead of updating numeric parameters, it operates in the space of natural language policies, providing an interpretable pathway from evaluation signals to prompt optimization.

## 4.4 Aggregated Policy Refinement Across Tasks

The process above generates numerous policy-editing suggestions across different tasks and trajectory segments. These suggestions are often redundant, overlapping, or sometimes contradictory. Applying them directly would lead to noisy and unstable prompt updates. To ensure robust refinement, TAVO includes an aggregation step. For each agent $a_i$, edits collected across all its involved trajectories and tasks are consolidated:

$$P_i^{\mathcal{T}} = \{P_{i,t}^{\mathcal{T}}\}_{t=1}^{T}, \tag{5}$$

$$\Delta P_i' = \text{TAVO}_{\text{aggregate}}(\{\Delta P_i^{\mathcal{T}}\}_{\mathcal{T} \in \mathcal{B}}), \tag{6}$$

where $\mathcal{B}$ denotes a batch of tasks. The aggregation module $\text{TAVO}_{\text{aggregate}}$ (implemented with an LLM) identifies common themes, merges redundant suggestions, and resolves conflicts by favoring coherent, high-impact edits. The final, refined prompt for agent $a_i$ is then produced by:

$$P_i^{\text{new}} = \text{TAVO}_{\text{refine}}(P_i^{\text{old}}, \Delta P_i'). \tag{7}$$

This aggregation transforms a collection of localized, potentially noisy edits into a stable and generalizable policy update, ensuring improvements are consistent across diverse tasks. Conceptually, this entire loop—trajectory rollout, credit assignment, suggestion generation, and aggregated refinement—parallels policy iteration in RL. TAVO adapts this powerful principle to the domain of verbalized policies, enabling systematic and interpretable optimization of LLM-based MAS.

## 4.5 Connections and Advantages of TAVO

TAVO is grounded in reinforcement learning (RL) principles but tailored to the constraints of LLM-based systems. Like RL, it builds on the MDP formalism and addresses credit assignment for iterative policy improvement, but it operates in the *verbalized policy space*, enabling use with frozen or gradient-inaccessible models—a common case for proprietary LLMs (see Table 1). Unlike standard RL, TAVO avoids dense reward design by using LLMs to generate interpretable credit signals. Compared to existing prompt optimization methods, it provides a more systematic trajectory-based decomposition, yielding three key benefits: (1) **Interpretability**: updates are expressed in natural language, ensuring transparency; (2) **Generalizability**: task-level aggregation supports transferable improvements; and (3) **Accessibility**: no gradient access is required.

## 5 Experimental Setup

**Benchmark and Evaluation.** We conduct the experiments on *MultiAgentBench* (Zhu et al., 2025), which is a comprehensive benchmark designed to evaluate LLM-based multi-agent systems. We conduct experiments on three collaborative domains: *Research*, *Coding*, and *Database*, and one competitive domain: *Bargaining*. For each domain, we create disjoint train/validation/test partitions

| Aspect | Traditional RL | TAVO | Example - TAVO |
|---|---|---|---|
| Policy | Numeric parameters (e.g., neural network weights) optimized via gradient-based methods. | Natural language prompts (e.g., behavior rules, or task guideline) optimized in verbalized (non-gradient) way | *Encourage comprehensive literature reviews and integration of recent findings into research proposal* |
| Optimization Signal | Numeric rewards, often sparse and delayed. | Verbalized feedback derived from trajectory-aware credit assignment and outcome evaluation. | *The task was completed thoroughly, addressing all specified requirements including literature review, brainstorming, summarization...* |
| Credit Assignment | Numeric proxy, like value estimation and advantage functions. | Verbalized process feedback derived from outcome feedback based on sub-trajectory. | *The current iteration makes significant progress towards task completion by defining the research question and developing a detailed methodology...* |
| Update Mechanism | Policy gradient derived from numeric credit. | Policy-editing suggestion derived from verbalized outcome and process feedback. | *Modify literature review prompts to ensure comprehensive coverage and integration of recent advancements* |

Table 1: Comparison between traditional reinforcement learning (RL) and our proposed TAVO.

| | Research | | | Coding | | | Database | | |
|---|---|---|---|---|---|---|---|---|---|
| | TS | CS | Milestones | TS | CS | Milestones | TS | CS | Milestones |
| Baseline | 80.00 | 84.69 | 5.55 | 51.00 | 63.26 | 7.93 | 47.33 | 89.33 | 9.90 |
| Reflexion | 79.33 | 86.10 | 5.95 | 49.33 | 63.04 | 6.53 | 44.67 | 91.00 | 10.00 |
| DsPy | 82.33 | 82.95 | 6.10 | 50.67 | 64.00 | 8.87 | 54.67 | 93.90 | 10.50 |
| TextGrad | 80.33 | 86.25 | 6.00 | 51.00 | 67.62 | 9.13 | 56.00 | 88.65 | 11.20 |
| TAVO | **83.00** | **88.50** | **7.50** | **52.00** | **85.54** | **9.87** | **58.67** | **94.00** | **11.80** |

Table 2: Task Score (TS), Coordinate Score (CS), and Average Milestone Achievement on three collaborative domains: *Research*, *Coding*, and *Database*.

| | TS | CS | Deal Ratio | Milestones | | |
|---|---|---|---|---|---|---|
| | | | | Average | per round | per 1M tokens |
| Baseline | 75.09 | 83.82 | 0.37 | 5.86 | 0.33 | 109.65 |
| Reflexion | 80.33 | 83.15 | 0.40 | 4.13 | 0.36 | 70.27 |
| DsPy | 55.00 | 78.38 | 0.35 | **10.71** | 0.34 | 69.46 |
| TextGrad | 63.67 | 78.33 | 0.50 | 8.80 | 0.33 | 52.75 |
| TAVO | **82.33** | **85.03** | **0.55** | 4.72 | **0.42** | **122.90** |

Table 3: Task Performance and Milestone on One Competitive Domain, i.e., *Bargaining*.

at the instance level. For evaluation, we follow the original benchmark and employ **Task Score**, **Coordination Score** and **Milestones** as the metrics. **Task Score** is computed to evaluate the final output quality. **Coordination Score** is computed to quantify the agents' communication and planning capabilities. As for **Milestone**, each task is segmented into a series of flexible milestones, and an LLM-based detector continuously monitors the iterative process to identify which milestones have been achieved. We report the average achieved milestone across tasks in the table. More details about the benchmark are in Appendix B.

**Baselines.** We compare TAVO with the following baselines: (1) Baseline: original prompts provided by the benchmark. (2) Reflexion (Shinn et al., 2023): which involves the verbalized feedback as part of the prompt. (3) DsPy (Khattab et al., 2024): which utilizes the numeric feedback to evolve prompts. (4) TextGrad (Yuksekgonul et al., 2025): which uses text gradient as a proxy to differentiate from the verbalized feedback.

**Model and Prompts.** Unless otherwise noted, all task-executing agents employ *gpt-4o-mini-2024-07-18* as the backbone model within the multi-agent system, while optimization and evaluation use *gpt-4o-2024-08-06*. For both TAVO and the baselines, we optimize prompts on the training set, select the best prompt on the validation set using milestone-based metrics, and report final results on the test set. Prompt optimization is limited to five rounds, with details provided in Appendix D.

## 6 RESULTS ANALYSIS

### 6.1 PERFORMANCE IN COLLABORATIVE AND COMPETITIVE SYSTEMS

We report results in collaborative and competitive tasks in Table 2 and Table 3, respectively.

| | Research - Milestone | | Coding - Milestone | | Database - Milestone | |
|---|---|---|---|---|---|---|
| | per round | per 1M tokens | per round | per 1M tokens | per round | per 1M tokens |
| Baseline | 1.21 | 16.42 | 2.07 | 93.02 | 1.98 | 79.80 |
| Reflexion | 1.22 | 17.24 | 2.02 | 84.95 | 2.00 | 63.29 |
| DsPy | 1.27 | 18.10 | 2.23 | 97.70 | 2.27 | 81.31 |
| TextGrad | 1.20 | 10.53 | 2.08 | 84.76 | 2.24 | 61.83 |
| TAVO | **1.50** | **20.40** | **2.43** | **110.29** | **2.36** | **91.58** |

Table 4: Milestone per Round and per Million Tokens Usage on Collaborative Domains.

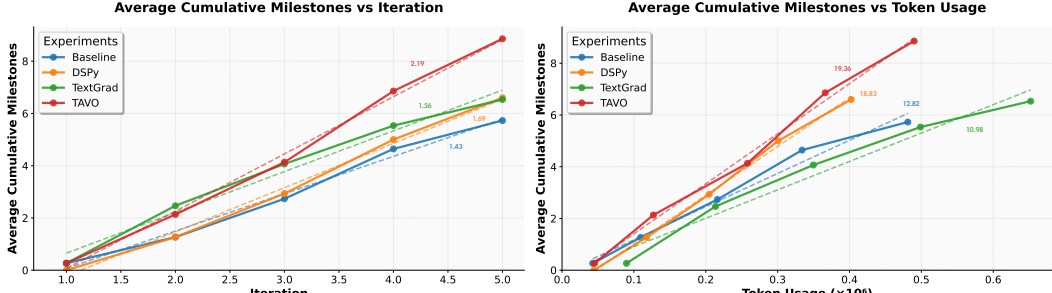

Figure 3: Efficiency Comparison between Different Methods in *Research*. The average cumulative milestone is showed based on the number of iteration (*left*) and token usage (*right*). The slope indicates the rate of achieving milestone, where the higher slope indicates the better efficiency.

**Our proposed TAVO consistently outperform the baselines across domains.** In three collaborative and one competitive domains, all prompt refinement methods see a consistent improvement on *Task Score* and *Coordinate Score*, which confirms the value of refining the prompt based on the suggestions from rollout. Moreover, our method TAVO noticeably outperforms two optimization methods solely on the outcome feedback (i.e., DsPy and TextGrad) across all domains. This confirms that incorporating trajectory information is crucial for optimizing a multi-agent system.

**The optimization of TAVO consistently enhances achievement on the milestone, leading to a better task performance.** To better understand the effectiveness of our TAVO, we compare *Milestone* achieved in the process in average. In collaborative domains, our TAVO achieves one more milestone when compared to other optimization methods that only utilize the outcome feedback. The only exception is in *Bargaining*, where DsPy and TextGrad increase the round of biding (i.e., milestone) but not help to reach a deal. Instead, our methods sees a decreased milestone but leading to higher deal ratio (More analysis in Appendix E). It demonstrates that the performance gain of our methods comes from the milestone achievement in the process. This further confirms the value of utilizing trajectory information in enhancing the coordination among agents in a multi-agent system to achieve an enhanced task performance.

**Trajectory Information is not only helpful to the effectiveness, but also to the efficiency.** We further show the milestone per round and per million tokens usage in Table 4 and Table 3. The results show that our TAVO achieves more milestones with the same budget, either per round or per million tokens usage. The observation demonstrates that the enhanced performance of our TAVO comes from the milestone achievement under fewer rounds or token usage. We further show the achievement progress of milestones under different rounds and token usage in Figure 3, where the slope shows the efficiency of the optimized methods. The results show that DsPy achieves more milestones with fewer token usage but with more rounds, while TextGrad achieves more milestones with fewer rounds but more token usage. Instead, our TAVO takes fewer tokens and fewer rounds to achieve a better performance. This confirms the value of involving trajectory information in enhancing the efficiency of a multi-agent system.

### 6.2 Effectiveness of Credit Assignment and Verbalized Policy

**Credit Assignment Mechanism is necessary to align the enhancement between milestone achievement and final outcome.** To further understand the effect of our *Credit Assignment Mechanism*, we remove credit assignment and only use the outcome feedback for generating suggestions

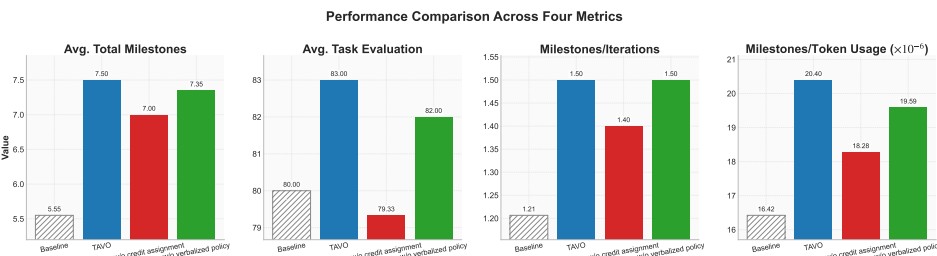

Figure 4: Ablation on local evaluation and verbalized policy in Research.

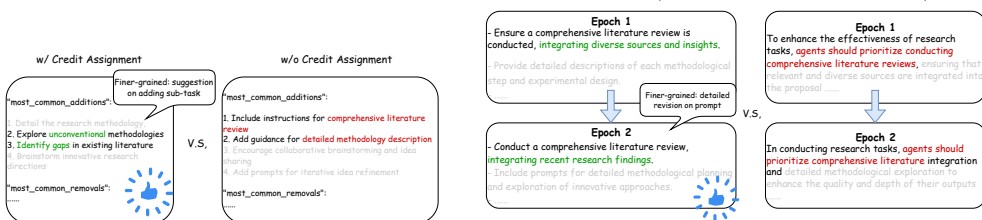

(a) Policy-Editing Suggestion Comparison    (b) Prompt Refinement Comparison.

Figure 5: Case Study for Ablation Study from *Research*. (a) Policy-editing suggestion comparison between w/ and w/o credit assignment, where credit assignment enables a finer-grained suggestion, e.g., "identify gaps". (b) Prompt refinement between w/ and w/o verbalized policy, where verbalized policy allows a finer-grained revision, e.g., "integrating recent research findings".

and refining prompts. As the results in Figure 4, the performance without credit assignment still outperforms the baselines except the task score. This demonstrates that optimizing without credit assignment can enhance the milestone achievement, but cannot lead to an enhancement in the final outcome. The example shown in Figure 5 demonstrates that the credit assignment enables the generation of finer-grained suggestions (e.g., identify gaps), leading to an improved achieved milestone and task score. This further confirms the necessity of our proposed *Credit Assignment Mechanism*.

**Verbalized Policy enables a more effective utilization of the trajectory-aware feedback.** To better understand the role of verbalized policies in leveraging trajectory-aware feedback, we remove policy verbalization. In this setting, the entire prompt is treated as a single unit, and we generate suggestions and refine the prompt as a whole. The results are reported in Figure 4, which demonstrates that the performance gain from trajectory information remains, but it sees a performance gap when compared to optimizing in a verbalized policy format. We further show one example in Figure 5 to indicate the effectiveness of verbalized policy. It can be observed that verbalized policy helps to perform the finer-grained revision, such as from "integrating diverse sources and insights" to a clear revision "integrating recent research findings". This observation confirms that verbalized policies help to utilize the fine-grained feedback captured from the trajectory.

## 7 CONCLUSION

In this work, we addressed the challenge of optimizing prompts for LLM-based multi-agent systems, where conventional methods that rely solely on outcome-based signals often fail to capture the nuances of agent interaction. We introduced TAVO, a Trajectory-Aware Verbalized Optimization framework that draws inspiration from reinforcement learning to create a more effective refinement process. By implementing a credit assignment mechanism, TAVO generates fine-grained, process-level feedback from interaction trajectories. We further conceptualize prompts as verbalized policies, allowing this rich feedback to be translated into concrete, actionable editing suggestions. Our experiments across both collaborative and competitive benchmarks demonstrate that TAVO not only improves final task outcomes and process milestones but also enhances coordination efficiency. These findings underscore the significant value of leveraging detailed trajectory data, paving the way for more adaptive, efficient, and robust multi-agent systems.

REPRODUCIBILITY STATEMENT

To aid reproducibility, we provide an anonymous link https://anonymous.4open.science/r/TAVO to the full source code. This repository encompasses all required scripts for data processing, model training, and evaluation, thereby allowing others to reproduce our results.

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

## A  METHOD DETAILS

We show the pseudo-code in Algorithm 1

---

**Algorithm 1** TAVO: Trajectory-Aware Verbalized Optimization

---

**Input:** Task set $\mathcal{B} = \{\mathcal{T}\}$, agents $\mathcal{A} = \{a_i\}_{i=1}^N$, initial prompts $P^{(0)} = \{P_i^{(0)}\}$, iterations $L$
**Output:** Refined prompts $P^{(L)}$

1: **for** $\ell = 1$ to $L$ **do**
2:    **for** each task $\mathcal{T} \in \mathcal{B}$ **do**
3:       $(r, \{h_t\}_{t=1}^T) \leftarrow \text{ROLLOUT}(\mathcal{T}, \mathcal{A}, P^{(\ell-1)})$
4:       $F_r \leftarrow R(\mathcal{T}, r)$             ▷ Feedback on outcome
5:       **for** $t = 1$ to $T$ **do**
6:          $F_t \leftarrow \text{TAVO}_{\text{credit}}(\mathcal{T}, h_t, F_r)$       ▷ Credit Assignment
7:          **for** each agent $a_i$ active in $h_t$ **do**
8:             $\Delta P_{i,t} \leftarrow \text{TAVO}_{\text{suggestion}}(\mathcal{T}, F_r, F_t, P_i^{(\ell-1)})$    ▷ Policy-Editing Suggestion
9:          **end for**
10:       **end for**
11:    **end for**
12:    **for** each agent $a_i$ **do**
13:       $\Delta P_i' \leftarrow \text{TAVO}_{\text{aggregate}}(\{\Delta P_i^{\mathcal{T}}\}_{\mathcal{T} \in \mathcal{B}})$       ▷ Suggestion Aggregation
14:       $P_i^{(\ell)} \leftarrow \text{TAVO}_{\text{refine}}(P_i^{(\ell-1)}, \Delta P_i')$         ▷ Prompt Refinement
15:    **end for**
16: **end for**
17: **return** $P^{(L)}$

---

## B  BENCHMARK DETAILS

The *Research* scenario involves multi-stage collaborative workflows spanning literature review, brainstorming, synthesis, and a 5Q-style proposal; *Coding* focuses on collaborative code generation and repair; *Database* emphasizes structured querying, read–write operations, and consistency reasoning; *Bargaining* evaluates multi-round negotiation where agents must maintain strategic coherence and reach valid agreements.

Regarding the evaluation, we follow the original benchmark and employ **Task Score**, **Coordination Score** and **Milestones** as the evaluation metrics. **Task Score** is computed to evaluate the final output quality. For tasks of **Research** and **Bargaining**, an LLM-defined scoring rubric is applied to generate the score. For other tasks, a rule-based metric is employed for evaluation. **Coordination Score** is computed to quantify the agents' communication and planning capabilities. As for **Milestone**, each task is segmented into a series of flexible milestones, and an LLM-based detector continuously monitors the iterative process to identify which milestones have been achieved. We report the average achieved milestone across tasks in the table. It is notable that **Task Score** focuses more on the quality of the final outcome, while **Milestone** provides a process-based evaluation.

For each domain, we create disjoint train/validation/test partitions at the instance level. In *Research* and *Bargaining* we use 5/5/20 splits, while *Coding* adopts 5/5/15 and Database uses 3/3/10, reflecting domain-specific task volumes.

## C  BASELINE

We compare our proposed TAVO with the following baselines:

1. Baseline: we use the original prompts provided by the benchmark as the baselines for comparison. Note that the prompt here is without any optimization.

2. Reflexion Shinn et al. (2023): this is a clasic method, which takes the verbalized feedback as the part of the prompt to construct the optimized prompt.

3. DsPy Khattab et al. (2024): this is the evolution-based method that utilizes the numeric feedback to filter the better prompts.

4. TextGrad Yuksekgonul et al. (2025): this is one of the recent methods that uses text gradient as a proxy to differentiate from the verbalized feedback, which is used for refining the prompt.

# D MODELS AND PROMPTS

## D.1 MODELS

Unless otherwise specified, all task-executing agents use *gpt-4o-mini-2024-07-18* as the backbone model to run multi-agent dialogues and produce task solutions. The optimizer that performs trajectory-aware feedback aggregation and policy editing uses *gpt-4o-2024-08-06*. This separation follows our design goal: maintain a lightweight, fast inference model for the agents while leveraging a stronger model to summarize trajectories and synthesize high-quality verbalized policies. For task score evaluation, we employ *gpt-4o-2024-08-06* as the evaluator. For both of our proposed TAVO and the used baselines, we optimize the prompts on the training set, select the best prompt on the validation set, and finally report the results on the held-out test set. For optimization on the training set, we run at most five optimization rounds. For selection on the validation set, we adopt milestone-based metrics as the primary selection criterion.

## D.2 PROMPTS

We list the used prompts in our experiments.

---

**Prompt for Trajectory-Aware Credit Assignment**

```
Please evaluate the quality and progress of the current iteration
    in this multi-agent collaboration.

Current Iteration Context:
Iteration Number: {iteration_number}
Iteration Content: {iteration_content}

Task Context:
Task Background: {task_context}
Previous Iterations: {previous_iterations}

Evaluation of final result:
{global_evaluation}

evaluation_criteria:

iteration_progress:
description: "Progress made in current iteration"
evaluation_points:
"Whether meaningful progress was made"
"Whether intermediate results are valuable"
"Whether the direction is correct and effective"

agent_coordination:
description: "Agent coordination in current iteration"
evaluation_points:
"Whether agents worked together effectively"
"Whether communication was clear and purposeful"
"Whether work division was appropriate"

stage_appropriateness:
```

---

```
description: "Evaluate whether the output meets the current stage"
evaluation_points:
"Whether the content depth is suitable for the current stage"
"Whether it is prepared for the subsequent work"
"Whether the time arrangement is reasonable"

output_format:
Please return the evaluation results in JSON format:
{
"iteration_progress": {
"score": score(1-10),
"analysis": "progress analysis",
"key_achievements": ["key achievement 1", "key achievement 2"]
},
"agent_coordination": {
"score": score(1-10),
"analysis": "coordination analysis",
"coordination_highlights": ["coordination highlight 1",
    "coordination highlight 2"],
"coordination_issues": ["coordination issue 1", "coordination
    issue 2"]
},
"stage_appropriateness": {
"score": score(1-10),
"analysis": "stage appropriateness analysis",
"alignment_evidence": ["alignment evidence 1", "alignment evidence
    2"]
},
"iteration_summary": {
"overall_score": average score,
"main_contributions": ["main contribution 1", "main contribution
    2"],
"areas_for_improvement": ["area for improvement 1", "area for
    improvement 2"],
"next_iteration_suggestions": ["next iteration suggestion 1",
    "next iteration suggestion 2"]
}
}
```

**Prompt for Suggestion-Aggregated Policy Refinement**

```
 You are given a list of short rules/suggestions (may be redundant
    or semantically similar).
Task:
(1) normalize/merge near-duplicates;
(2) rank by (estimated frequency + practical importance);
(3) return the top-{top_k} representative and concise items.

Input items (one per line with index):
{input_items}

Return STRICT JSON only:
{{
  "top": [
    {{
      "text": "representative concise rule",
      "support_examples_idx": [1,5,9],
      "support_count": 3,
```

```
      "importance": 0
    }}
  ]
}}
```

**Prompt for Generating Policy-Editing Suggestions**

```
 Please analyze the following Agent's performance and provide
    improvement recommendations based on final result quality:

**Task Context:**
{task_context}

**Final Result Quality Assessment:**
{result_context}

**Full Result Evaluation:**
{result_evaluation}

**Agent Information:**
- Agent ID: {agent_history['agent_id']}
- Agent Type: {agent_history['agent_type']}
- Profile: {agent_history['profile']}

**Current System Prompt:**
{agent_history['original_system_prompt']}

**Agent Historical Performance:**
- Number of tasks executed: {len(agent_history['tasks_performed'])}
- Number of communications: {len(agent_history['communications'])}
- Token usage: {agent_history['token_usage']}

**Detailed Task History:**
{agent_history['tasks_performed']}...

**Communication History:**
{agent_history['communications']}...

**Result History:**
{agent_history['results']}...

**Result Quality-Based Deep Analysis Requirements:**

1. **Result-Oriented Assessment**: Analyze whether this agent's
    contributions are effective based on final result quality
2. **Causality Analysis**: Analyze the causal relationship between
    this agent's behavior and final result quality
3. **Impact Verification**: Whether this agent's work had positive
    impact on final results
4. **Problem Attribution**: If result quality is poor, whether
    this agent is one of the influencing factors
5. **Targeted Improvement**: Provide targeted improvement
    recommendations based on result quality issues

**Important**: Please judge agent performance effectiveness by
    combining result quality assessment, not just process analysis.

Please return evaluation results in JSON format:
```

```
{{
    "overall_score": 1-10,
    "result_oriented_analysis": {{
        "contribution_to_final_result": "Analysis of this agent's
specific contribution to final result",
        "effectiveness_rating": 1-10,
        "impact_on_quality": "Analysis of impact on result quality"
    }},
    "strengths": ["Strength 1 based on result verification",
"Strength 2 based on result verification", ...],
    "weaknesses": ["Weakness 1 affecting result quality",
"Weakness 2 affecting result quality", ...],
    "causality_analysis": {{
        "positive_contributions": ["Behavior 1 promoting result
quality", "Behavior 2 promoting result quality"],
        "negative_impacts": ["Behavior 1 damaging result quality",
"Behavior 2 damaging result quality"],
        "missed_opportunities": ["Missed opportunity 1 to improve
results", "Missed opportunity 2 to improve results"]
    }},
    "prompt_suggestions": {{
        "result_oriented_improvements": "Prompt improvement
suggestions based on result quality issues",
        "effectiveness_enhancements": "Prompt modifications to
improve actual effectiveness",
        "quality_focus_additions": "Prompt additions to enhance
result quality awareness",
        "collaboration_optimizations": "Prompt adjustments to
optimize collaboration effects"
    }},
    "targeted_recommendations": {{
        "immediate_fixes": ["Immediate improvement 1", "Immediate
improvement 2"],
        "strategic_improvements": ["Strategic improvement 1",
"Strategic improvement 2"],
        "quality_assurance_measures": ["Quality assurance measure
1", "Quality assurance measure 2"]
    }},
    "specific_prompt_modifications": {{
        "add_instructions": ["Specific instruction 1 to add",
"Specific instruction 2 to add"],
        "remove_content": ["Content 1 to remove", "Content 2 to
remove"],
        "restructure_suggestions": ["The content to be modified
1(and the way to modify)", "The content to be modified 2(and
the way to modify)"]
    }}
}}
```

## E  EVALUATION METRICS ON COMPETITIVE TASK: BARGAINING

The bargaining scenario is competitive: a buyer and a seller exchange proposals and counterof-
fers, and progress may involve strategic hesitation or cycling. In this setting, average milestone
counts alone are not diagnostic, because intermediate events (e.g., offer/counteroffer) can accumu-
late without reflecting substantive movement toward agreement. To account for this difference from
cooperative tasks, we place greater emphasis on outcome-oriented metrics—deal closure rate and
task/coordinate scores—while also reporting efficiency indicators such as Average Milestones per
round and per 1M tokens to characterize system throughput. Our evaluation thus balances effec-

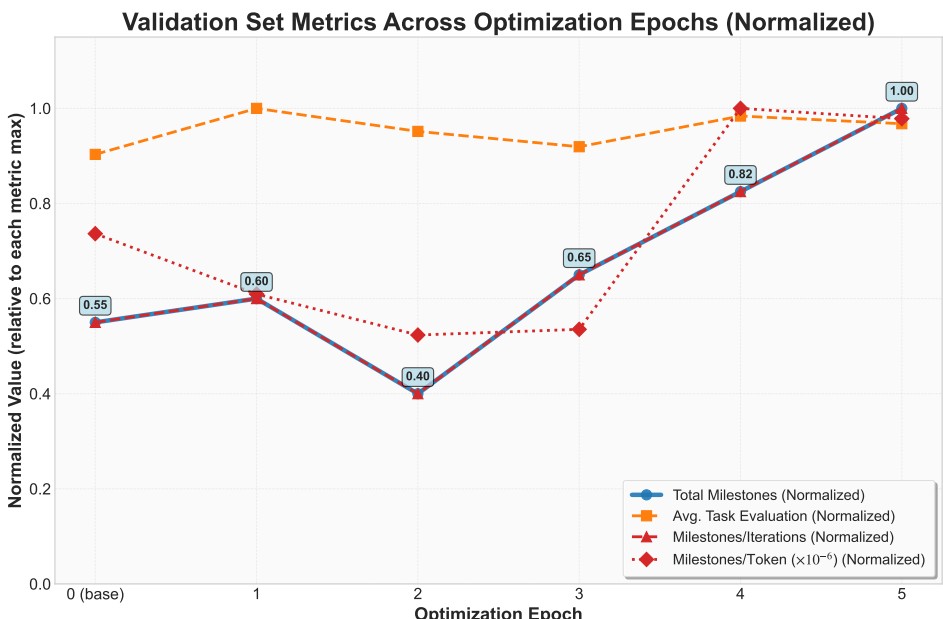

Figure 6: Validation results across five optimization epochs.

tiveness (whether and how well agreements are reached) with operational efficiency, rather than interpreting the absolute value of average milestones as a standalone measure of performance.

## F    MORE RESULTS

We report four indicators on the held-out set: overall throughput (Total Milestones), outcome quality (Avg. Task Evaluation), and two efficiency proxies (Milestones per iteration and per million tokens). Across five iterations, we observe a non-monotonic but interpretable pattern. Task Evaluation improves from the base to iteration 1 (74.7→82.7) and remains competitive thereafter with mild fluctuations (76–82.7). In contrast, efficiency and throughput rise more substantially in later epochs: Milestones/Iterations increases from 0.88 (base) to 1.60 (iter 5), and Total Milestones grows steadily ($\approx$4.4→8.0). Milestones/Token decreases early (22.54→16.02 by iter 2), then rebounds to a higher level by iter 4–5 ($\approx$30), indicating a shift toward more aggressive milestone production per compute.

These trends suggest an evolving balance between effectiveness and operational efficiency. Early iterations likely favor conservative, quality-preserving behaviors (higher Task Evaluation with tighter token budgets), while later iterations emphasize faster progress and higher milestone density. The modest softening of Task Evaluation at iterations 3–5, alongside strong gains in Milestones/Iterations and Total Milestones, is consistent with a throughput-oriented policy that does not always translate additional steps into proportional quality gains.

## G    THE USE OF LARGE LANGUAGE MODELS (LLMS)

This work was developed with the support of Large Language Models (LLMs), which served as a writing assistant and a key component in our experiments. Specifically, ChatGPT aided in drafting and refining the text, enhancing the clarity and precision of our expression, while LLMs were also actively employed in the experimental pipeline.

