# OpenReview forum: "Trajectory-Aware Verbalized Optimization for Multi-Agent Systems"
_ICLR.cc/2026/Conference — ICLR 2026 Conference Withdrawn Submission_

### Official Review · Reviewer_CUQx · 2025-10-20

**Soundness:** 3
**Presentation:** 3
**Contribution:** 2
**Rating:** 2
**Confidence:** 4

**Summary:**

This paper presents TAVO, a framework for optimizing prompts in multi-agent systems by leveraging trajectory-level information. The method formalizes prompts as verbalized policies and introduces a trajectory-aware credit assignment mechanism that decomposes long-horizon interactions into sub-trajectories. Process-level feedback is verbalized and used to refine prompts iteratively. Experiments on MultiAgentBench are reported across collaborative and competitive settings.

**Strengths:**

1.Step-level error attribution for multi-agent systems is an important problem.

2.It makes sense to use LLMs to conduct credit assignment.

**Weaknesses:**

1. The proposed "trajectory-level credit assignment" approach is highly similar to reflective frameworks such as Reflexion[1], which both use interactive trajectories and feedback from final outcomes for self-correction. This paper refines this approach only by explicitly partitioning trajectories into sub-trajectories, representing an engineering improvement rather than a fundamental innovation.
2. The paper claims to extend verbal-feedback paradigms to multi-agent settings with role-aware policies (L135). However, it remains unclear what new challenges arise from this extension or how the proposed framework addresses them technically.
3. The method largely assumes fixed communication topologies and dependency structures, overlooking the causal and dynamic nature of multi-agent optimization (cf. GPT-Swarm [2]).
4. The horizontal comparison in Table 1 does not convincingly highlight the novelty of TAVO. Claimed features such as “interpretability,” “trajectory-level reasoning,” and “gradient-free optimization” are already present in prior prompt optimization frameworks.
5. The paper attributes “interpretability, generalizability, and accessibility” to TAVO’s advantages over existing prompt optimization methods (L315–317). However, interpretability and accessibility are not unique to TAVO.
6. All experiments are conducted on MultiAgentBench and have not been validated on different benchmarks (e.g., GAIA, AgentBench, or complex reasoning environments). Furthermore, the baseline appears to only select single-agent prompt optimization methods and does not include multi-agent optimization frameworks such as GPT-Swarm
7. The authors claim that “outcome feedback improves process-level milestones but not final performance (L458–460),” yet provide no explanation of why such asymmetry arises. A causal or diagnostic analysis would strengthen the argument.
8. Key modules—such as the policy-editing suggestions (Sec 4.3)  and aggregated policy refinement (Sec 4.4) components—are not individually ablated, making it difficult to assess their specific contributions.
9. The paper does not test whether the approach remains effective under changes in the base LLM or initial prompt settings.
10. Minor issues. L476–477: “we addressed, we introduced” → “we address, we introduce” .
11. How does TAVO achieve adaptivity and robustness? (L485)

[1] Reflexion: Language Agents with Verbal Reinforcement Learning, NIPS23

[2] GPTSwarm: Language Agents as Optimizable Graphs, ICML24

**Questions:**

1. How can a rule-based evaluator generate natural-language feedback that is comparable to that produced by an LLM-as-a-Judge? (L252)
2. What are the specific sizes of the training, validation, and test sets, and how many tasks are included in each?
3. Does Table 4 report the inference cost? If so, what are the token or time costs associated with the optimization phase, such as trajectory decomposition, feedback generation, and aggregation

---

### Official Review · Reviewer_oQV8 · 2025-10-31

**Soundness:** 2
**Presentation:** 2
**Contribution:** 2
**Rating:** 4
**Confidence:** 4

**Summary:**

This work proposes a Trajectory-Aware Verbalized Optimization framework, TAVO, which incorporates a trajectory-aware credit assignment mechanism and conducts trajectory-level textual gradients for improving the prompts per agent. TAVO consists of a multi-agent workflow for credit assignment, editing suggestions, prompt aggregation, and refinements. TAVO is validated on the MultiAgentBench with the backbone model as GPT-4o-mini, where TAVO shows improvements over the baseline methods: Reflexion, DSPy, and TextGrad.

**Strengths:**

1. Trajectory-level credit assignment is an important research topic in the field of multi-agent optimization, and TAVO enhances previous textual gradient style methods to a more fine-grained level for verbalized optimization.

2. The algorithm itself (Algorithm 1) is clear, and the logic makes sense.

3. The performance improvement in the MultiAgentBench is clear.

**Weaknesses:**

1. One major limitation of this framework is that all prompts of the LLM-optimizers in TAVO, including the credit assignment agent, have been highly manually engineered (See Appendix D.2), therefore brittle, and appeared to overfit to the task of MultiAgentBench. This raises the concern of the generalization of the framework to other general domains of tasks. In addition, I do not find the proposed approach fundamentally different to the TextGrad methods.

2. Similarly, the evaluation is only conducted on the MultiAgentBench and on a single family of LLM backbone (i.e., GPT4o-mini). It is important to verify the proposed approach in the general domain of reasoning benchmarks, such as SuperGPQA, BBEH, for further validating the effectiveness of the framework.

3. Though I am generally in favor of the credit assignment idea to improve the existing textual gradient prompt optimization approaches, there are many reasoning tasks, such as Math, where the middle stage of reasoning misses the ground truth. Consequently, it is difficult to analyze the faithfulness of intermediate reasoning chains, where the credit assignment is also likely to fail there. The TAVO framework highly relies on a single LLM-optimizer to analyze intermediate reasoning results, which may not be reliable in many scenarios.

4. The implementation details of the baselines are unclear to me. I think the TextGrad is the strongest baseline there, but how is TextGrad implemented in the multi-agent context? The missing implementation details of baselines make the performance improvements by TAVO less convincing. Other automatic multi-agent prompt optimization approaches should be considered, such as MIPROv2 [1] and the symbolic optimizer [2].

Some minor points: I find the connection of TAVO to RL is weak. The credit assignment is a general idea in optimization problems for multi-agent systems. It may be better not to refer to RL that many times in writing.

[1] Opsahl-Ong, K., Ryan, M. J., Purtell, J., Broman, D., Potts, C., Zaharia, M., & Khattab, O. (2024). Optimizing Instructions and Demonstrations for Multi-Stage Language Model Programs (No. arXiv:2406.11695). arXiv. http://arxiv.org/abs/2406.11695

[2] Zhou, W., Ou, Y., Ding, S., Li, L., Wu, J., Wang, T., Chen, J., Wang, S., Xu, X., Zhang, N., Chen, H., & Jiang, Y. E. (2024). Symbolic Learning Enables Self-Evolving Agents (No. arXiv:2406.18532). arXiv. http://arxiv.org/abs/2406.18532

**Questions:**

1. How reliable is the credit assignment agent in the general reasoning domain?

2. How do you merge noisy prompt improvement suggestions? Is it guaranteed to resolve the conflicts? Sometimes, the suggestion may not lead to higher performance. How do you handle these scenarios?

3. Can you clarify the implementation of TextGrad to the multi-agent context? I think the TextGrad approach can also progressively assign credits to each agent and make verbalized policy improvements.

4. Can you provide some examples of the prompts before and after TAVO optimization?

---

### Official Review · Reviewer_k5h6 · 2025-10-31

**Soundness:** 3
**Presentation:** 2
**Contribution:** 2
**Rating:** 2
**Confidence:** 4

**Summary:**

Existing automated optimization methods for multi-agent systems largely rely on coarse-grained, task-level outcomes and do not fully exploit the rich trajectory-level information. This paper proposes Trajectory-Aware Verbalized Optimization (TAVO), a framework for prompt optimization in multi-agent systems. The framework introduces a credit-assignment mechanism that decomposes interaction trajectories into sub-trajectories and uses LLM-based reasoning to analyze which steps contributed to or hindered success, thereby generating fine-grained, process-level feedback. Experimental results demonstrate the effectiveness of the proposed method.

**Strengths:**

1. The proposed TAVO framework fully leverages trajectory-level signals, alleviating the sparsity and coarse-grained feedback issues in multi-agent optimization. Compared with task-level outcomes, it can precisely locate points of success and failure, improving optimization efficiency.
2. TAVO converts task rewards directly into concrete prompt-editing instructions, requiring no changes to the base model’s parameters—making it practical and friendly to closed-source models.
3. The paper is clearly written and easy to follow, and the experimental results validate the effectiveness of the proposed method.

**Weaknesses:**

1. The core of the method remains “reflection-based prompt optimization,” with the main novelty being the refinement of outcome-level signals into sub-trajectory feedback and their verbalization into prompt items. Compared with existing multi-agent reflection/prompt-optimization work, the contribution is primarily a granularity upgrade.
2. Credit assignment relies on outcome-level verbal evaluations, and the generation of suggestions also depends on stronger LLMs.Therefore, the proposed approach may be highly sensitive to the capabilityof the evaluator.
3. The method assumes prompts can be decomposed into mappable items and that sub-trajectory contributions can be robustly aligned to specific items. However, prompts are often semantically entangled with fuzzy boundaries, which may affect the stability and generalization of suggestion aggregation.
4. Although experiments are conducted on benchmarks across multiple scenarios, the performance gains are limited.

**Questions:**

See weaknesses.

**Details Of Ethics Concerns:**

The paper raises no ethical concerns.

---

### Official Review · Reviewer_Wtbv · 2025-11-01

**Soundness:** 2
**Presentation:** 3
**Contribution:** 2
**Rating:** 2
**Confidence:** 4

**Summary:**

This paper proposes TAVO (Trajectory-Aware Verbalized Optimization), a framework for prompt optimization in LLM-based multi-agent systems that integrates trajectory-level analysis into the optimization loop.

Traditional prompt optimization methods rely on outcome-only feedback, overlooking the reasoning and coordination processes that occur during agent interaction. TAVO addresses this by introducing two main innovations:
	1.	Trajectory-Aware Credit Assignment — Decomposes full interaction trajectories into sub-trajectories, generating fine-grained, verbalized feedback on process quality rather than only task outcomes.
	2.	Verbalized Policy Refinement — Treats prompts as verbalized policies, translating trajectory feedback into interpretable, natural-language editing suggestions, aggregated across tasks for stable, generalizable improvements.

TAVO is implemented using LLMs both as evaluators and as optimizers, employing an iterative optimization loop akin to policy iteration in reinforcement learning, but entirely in the space of natural language prompts.

Experiments on MultiAgentBench (across collaborative tasks like Research, Coding, and Database, and a competitive Bargaining task) demonstrate that TAVO achieves higher task scores, better coordination, and improved efficiency compared with outcome-based baselines such as DsPy, Reflexion, and TextGrad.

Ablation studies show that removing credit assignment or verbalized policy structure degrades performance, confirming the importance of both components.

**Strengths:**

The paper introduces a new optimization perspective by treating prompts as verbalized policies and modeling optimization as a form of verbalized policy iteration grounded in RL theory.

1. The analogy to reinforcement learning—credit assignment, value decomposition, and policy updates—is well-motivated and consistently developed throughout the paper.

2. For Trajectory-Level Analysis:

(1) TAVO effectively leverages process-level signals from trajectories, bridging a major gap between human expert diagnosis and automated optimization.

(2) The decomposition of long trajectories into sub-sequences and localized credit assignment improves interpretability and efficiency.

3. Empirical Evidence:

Results on four distinct domains (Research, Coding, Database, Bargaining) show clear and consistent gains in Task Score, Coordination Score, and Milestone metrics over all baselines. Efficiency analysis (milestones per iteration/token) demonstrates that TAVO achieves higher performance with fewer tokens and rounds, validating its practical scalability.

4. Interpretability and Reproducibility

(1) All optimization steps are interpretable (natural-language feedback and edits).

(2) The paper provides detailed algorithmic descriptions (Algorithm 1), prompt templates, and an open-source link for reproducibility.

**Weaknesses:**

1. Limited Theoretical Analysis

(1) While the RL analogy is conceptually elegant, the framework lacks formal theoretical guarantees (e.g., convergence or error bounds) for verbalized policy updates.

(2) The method is largely heuristic—driven by LLM-based evaluations rather than provable optimization properties.

2. Evaluation Bias from LLM-as-a-Judge

(1) The experiments rely on LLM-based evaluators (gpt-4o) for both optimization and scoring. This introduces potential evaluation leakage or bias, as the same or similar models both generate and assess outputs.

3. Absence of Cost or Stability Analysis

(1) No detailed analysis of computational cost, runtime, or variance across optimization rounds is provided.

(2) While Table 4 and Figure 3 suggest efficiency gains, the paper lacks a quantitative trade-off analysis between performance improvement and computation overhead.

4. Benchmark Scope and Diversity

(1) Experiments are limited to MultiAgentBench. While it includes collaborative and competitive tasks, further testing on open-ended or real-world benchmarks (e.g., web-based agents, embodied tasks) would better validate generalization.

5. Ablation Depth

(1) The ablations mainly show performance drops when removing components, but no analysis of failure cases or qualitative breakdowns of trajectory feedback quality is provided.

**Questions:**

1. How robust is TAVO to noisy or misleading trajectory feedback?
Since the credit assignment depends on LLM-generated verbal evaluations, how does the system handle inconsistent or incorrect local assessments?
2. Can TAVO generalize across unseen domains or agent configurations?
The framework aggregates task-specific edits—does it retain generality, or risk overfitting to the domains in MultiAgentBench?
3. How is aggregation stability ensured?
When merging multiple verbalized suggestions (Eq. 6), how does the model prevent oscillations or contradictory edits between optimization epochs?
4. How costly is the iterative optimization loop?
Each epoch involves multiple rollouts, evaluations, and refinement passes. What are the computational or token costs relative to baselines?
5. Could TAVO be extended to multimodal or embodied multi-agent tasks?
The paper focuses on text-based systems—would trajectory-aware verbalization still work when trajectories include visual or action sequences?

---

### Note · Authors · 2026-01-16

I have read and agree with the venue's withdrawal policy on behalf of myself and my co-authors.